# A Nuclear Belt Fastens on Neural Cell Fate

**DOI:** 10.3390/cells11111761

**Published:** 2022-05-27

**Authors:** Ivan Mestres, Judith Houtman, Federico Calegari, Tomohisa Toda

**Affiliations:** 1Center for Molecular and Cellular Bioengineering (CMCB), DFG-Center for Regenerative Therapies Dresden, Technische Universität Dresden, Fetscherstr. 105, 01307 Dresden, Germany; ivan.mestres_lascano@tu-dresden.de; 2Nuclear Architecture in Neural Plasticity and Aging, German Center for Neurodegenerative Diseases (DZNE), Tatzberg 41, 01307 Dresden, Germany; judith.houtman@dzne.de

**Keywords:** nuclear envelope, neurogenesis, chromatin regulation, neuronal migration, development, aging

## Abstract

Successful embryonic and adult neurogenesis require proliferating neural stem and progenitor cells that are intrinsically and extrinsically guided into a neuronal fate. In turn, migration of new-born neurons underlies the complex cytoarchitecture of the brain. Proliferation and migration are therefore essential for brain development, homeostasis and function in adulthood. Among several tightly regulated processes involved in brain formation and function, recent evidence points to the nuclear envelope (NE) and NE-associated components as critical new contributors. Classically, the NE was thought to merely represent a barrier mediating selective exchange between the cytoplasm and nucleoplasm. However, research over the past two decades has highlighted more sophisticated and diverse roles for NE components in progenitor fate choice and migration of their progeny by tuning gene expression via interactions with chromatin, transcription factors and epigenetic factors. Defects in NE components lead to neurodevelopmental impairments, whereas age-related changes in NE components are proposed to influence neurodegenerative diseases. Thus, understanding the roles of NE components in brain development, maintenance and aging is likely to reveal new pathophysiological mechanisms for intervention. Here, we review recent findings for the previously underrepresented contribution of the NE in neuronal commitment and migration, and envision future avenues for investigation.

## 1. Introduction

Neural stem cells (NSCs), like any other somatic stem cell, are characterized by their intrinsic ability to symmetrically proliferate; that is to give rise to more NSCs as well as to generating more differentiated progeny, including neurons and glia. Critically, controlling the balance between proliferation versus differentiation has a central role in the evolution, development, homeostasis and repair of the nervous system. Moreover, depending on the species, neurogenic niche and age of the organism being considered, a variety of stem and progenitor cell types can be found that are characterized by a different transcriptional signature, proliferative rate, lineage, morphology and other features [1,2,3,4,5,6]. After extensive research, we now understand a plethora of mechanisms governing NSC behavior and the fate commitment of daughter cells. Among others, recent reviews have already summarized our current understanding about the roles of intrinsic genetic programs, extracellular signals, cell adhesion, cell cycle dynamics, polarity and (a)symmetric cell division in regulating neurogenesis [1,2,3,4,5,6]. In addition, in the last decade an increasing number of studies have begun to highlight how the nuclear envelope (NE) can contribute to controlling the switch from proliferation to differentiation of NSCs during development and homeostasis of the nervous system. Far from the original concept that the NE merely represents a barrier separating the nucleo- from the cytoplasm, more sophisticated functions of the NE have recently been proposed, including providing a mechanical scaffold for both nuclear and cellular movement, as well as regulating chromatin architecture and gene expression, processes that are critical for controlling NSC fate. We will briefly explain the processes of the vertebrate central nervous system that involve NE components, and subsequently discuss the contribution of the NE to the control of those processes.

During development of the vertebrate central nervous system, NSCs form a pseudostratified epithelium where they reposition their nucleus as a function of their cell cycle phase in a process known as interkinetic nuclear migration (INM) [1] (Figure 1). Through INM, NSCs undergo S phase at more basal positions of the neuroepithelium, whereas mitoses occur at its apical, ventricular-facing surface. Hence, polarity cues that localize at the apical side of neuroepithelial cells can be inherited symmetrically or asymmetrically among their daughters, giving rise to proliferative or differentiative divisions, respectively [2,7,8]. As new-born neurons originate from NSCs, they migrate from the germinal zones to their final destination via locomotion and somal translocation [9,10] (Figure 1). As they migrate, later-born neurons need to surpass their predecessors, so-called birth-timing dependent inside-out migration. In addition, interneurons populate the cortical layers after tangential migration from their site of origin in sub-cortical areas [11]. This coordinated migration is the basis of proper brain development, and a failure of this process leads to severe deficits in brain function.

Critically, although the nucleus is the largest organelle in a cell, it needs to be propelled during INM and neuronal migration to successfully develop the central nervous system. This task requires an adaptable NE: tight enough not to break when pulled or pushed, but sufficiently flexible to deform when squeezing between neighboring cells without damaging the genetic material inside the nucleus. We will summarize how defects in NE components can cause failures in migration as well as damage to the genetic material.

In addition to neurogenesis during development, NSCs remain in the brain over the course of life and generate new neurons in adulthood. To date, NSC populations have been characterized in the subventricular zone (SVZ), dentate gyrus (DG), hypothalamus and spinal cord in mammals. This so called “adult neurogenesis” entails similar processes, including proliferation of NSCs and migration of their progeny [12]. Adult neurogenesis plays a prominent role in cognitive processes, since adult-born neurons migrate and are integrated into existing circuits to provide structural and functional plasticity, which contributes to cognitive flexibility and mood regulation [13,14]. Although adult NSCs are maintained over the course of life, there is a substantial decline in NSC function with increasing age [15]. Furthermore, several neurodegenerative disease models are associated with a reduction in adult neurogenesis [16], and various deficits in the NE have been implicated in neurodegenerative diseases. Thus, the NE is likely to play prominent roles not only during brain development, but also in the maintenance of brain function. We will summarize potential contributions of the NE to adult brain plasticity mainly focusing on adult neurogenesis.

In addition to migration, the NE determines chromatin organization and gene regulation during neural lineage specification and maintenance. Upon lineage specification of NSCs or migration of new-born neurons, specific genetic loci are silenced or activated following cell type-specific molecular programs [6]. In this process, silent genes are mainly located within compacted heterochromatin that is often associated with the NE, whereas active genes are often located in open euchromatin at the nuclear center. The topological switch between these two chromatin states can either hinder or facilitate gene transcription according to cellular requirements, and the NE supports robust gene repression by tethering specific heterochromatin regions in a cell type-dependent manner [17]. Given its functions in migration and the control of gene expression, it is hardly surprising that the NE is ideally suited to regulating NSC fate. Yet, to the best of our knowledge, only a handful of studies have directly addressed this aspect of the NE.

In order to fulfill the described roles in migration, chromatin organization and gene regulation, the NE consists of three parts. It comprises (1) the nuclear membrane separating the nucleus from the cytoplasm, (2) a nuclear lamina to provide physical support, and (3) nuclear pore complexes that mediate macromolecule interchange (Figure 2). In addition, the nuclear lamina and nuclear pore complexes also participate in chromatin organization and gene regulation (Figure 2). We will briefly explain the constitution of each of these components.

Lipid extractions from human nuclear membranes revealed an enrichment of phosphatidylcholines, phosphatidylinositides and cholesterol. This mixture grants the nuclear membrane an elasticity several times greater than that of the plasma membrane [18]. Additionally, the composition of nuclear gangliosides, which are the main glycosphingolipids in the brain, has been shown to change throughout development [19]. Whether or not lipid composition has a role in cell fate or migration remains to be clarified. Concerning protein composition, the most prominent component of the nuclear lamina is the type V intermediate filaments called lamins. There are four lamins found in mammalian somatic cells: lamins A and C are isoforms of the *lmna* gene, and lamin B1 and B2 are encoded by the independent genes *lmnb1* and *lmnb2*, respectively. Additionally, lamin C2 and B3 are isoforms of the *lmna* and *lmnb2* genes, respectively, and their expression is restricted to germ cells [20]. Lamins can be farnesylated at a CaaX motif, and through this modification, they are anchored to the inner nuclear membrane [21]. Additionally, lamin-associated proteins provide a stable physical coupling between the nucleus and the cytoskeleton. The nuclear pore complex (NPC) is a large assembly of eight spokes that combine to form a pore consisting of nearly 500 proteins named nucleoporins (Nups) [22,23,24]. These complexes regulate active transport of large molecules by interacting with nuclear transport receptors. Interestingly, super resolution microscopy and proteomics analyses showed that protein levels of nucleoporins vary according to cell type as well as in disease conditions, suggesting that NPC stoichiometry is adapted to cellular needs [25,26].

In this review, we will discuss NE components and the mechanisms regulating neural cell fate. We first describe their roles during neuronal lineage commitment, emphasizing how mutations may lead to organogenesis defects through their contribution to the biomechanical properties of the NE (Section 2) or their participation in gene and chromatin regulation (Section 3). We then discuss key aspects of the NE shown to influence cellular aging and neurodegenerative diseases (Section 4).

## 2. Nuclear Envelope during Nervous System Development and Maintenance

### 2.1. Nuclear Envelope and NSC Proliferation versus Differentiation

Cell division requires several adaptations in the NE. At the onset of prophase, cyclin-dependent kinase 1 (Cdk1, in complex with cyclin B), together with other kinases, hyperphosphorylate Nups first, and subsequently, lamins. Their phosphorylation at multiple sites degrades the connections between each other and with the DNA [27]. Thus, the nuclear pore complexes are disassembled, causing fenestration of the NE. The loosening of the NE allows spindle microtubules to access the chromosome kinetochores. By the end of mitosis, immediately after the segregation of duplicated chromatids, ESCRT-III (endosomal sorting complex required for transport III) coordinates reassembly of the NE. ESCRT-III associates with spastin, to sever the spindle microtubules, and with CHMP7 (charged multivesicular body protein 7), for membrane sealing. Once the NPCs are re-inserted in the NE, export of cytoplasmic constituents out of the nucleoplasm begins [27,28].

Disruption of NE disassembly and reassembly is known to be related to several disorders [29], including neurodevelopmental defects. As one example, upon neural progenitor cell entry into prophase, Cdk5 phosphorylates nudE neurodevelopment protein 1 like 1 (Ndel1) around the NE. This stimulates Ndel1 binding to lyssencephaly-1 (Lis1) and dynein, a microtubule motor protein involved in force generation. Together, Lis1 and dynein assist in disassembly of the NE by promoting membrane tearing (Figure 2). In haploinsufficient Lis1 mice, an increased number of dividing cells show an intact NE that halts them in prophase, indicating that Lis1 is necessary for NE break down and mitosis progression. These events have been proposed to favor asymmetric cell division and neurogenic commitment at the expense of proliferation, presumably through a block in mitosis. In turn, the neural progenitor pool is exhausted, which ultimately contributes to lissencephaly [30].

More recently, it has been proposed that the levels of lamin B1 and B2 can also influence the fate of NSCs. Studies using single and double knockout mouse mutants revealed that both lamin B types, while dispensable for gene regulation and proliferation of pluripotent embryonic stem cells, instead play major roles in the regulation of somatic stem cell fate during organogenesis [31] (see Section 3 for NE regulation of chromatin structure and cell fate). Consistently, lamin B1/B2 knock-out mice displayed several developmental defects, including a disorganized brain cortex, fewer alveoli in their lungs, impaired ossification and shorter bones, resulting in lethality soon after birth [31,32]. Specifically for cortical malformations, lamin B1/B2 depletion was proposed to cause defects in NSC spindle orientation, impaired INM and neuronal migration [31]. Although not directly demonstrated, these phenotypes are thought to reflect defects in the dynamics between the nucleoskeleton and cytoskeleton resulting from the interaction of lamin B1 and B2 with Ndel1. In addition, lamin B1 depletion was reported to favor astrogliogenesis at the expense of neurogenesis in both NSCs in culture and in the mouse embryonic cortex; conversely, lamin B1 overexpression stimulated neuronal differentiation [33].

Seemingly contrasting the astrogliogenic effect reported after lamin B1 knockdown during embryonic development [33], conditional knockout of lamin B1 in NSCs of the adult hippocampus led to premature neuronal differentiation resulting in long term exhaustion of NSCs [34]. Conversely, overexpression of lamin B1 repressed neuronal differentiation and promoted NSC proliferation in the adult hippocampus. Notably, these effects are consistent with the reported reduction in the levels of lamin B1 in NSCs during physiological aging, and is proposed to be one of the underlying causes of the decline in adult hippocampal neurogenesis with age [34,35]. These data suggest that lamin B1 may have different roles depending on the developmental context (see Section 3 for more detail about adult neurogenesis).

Additional examples for the contribution of the NE to stem cell differentiation also include proteins associated with the nuclear lamina, whose relevance is likely not limited to mammalian cortical development and adult neurogenesis. For instance, during zebrafish retina development, it has been pointed out that the outer nuclear membrane protein nesprin-2 serves as a nuclear anchor for cytoplasmic motor proteins, which is essential for propelling the nucleus during INM (Figure 2). The overexpression of the KASH (klarsicht/ANC-1/SYNE homology) domain of nesprin 2 functions in a dominant negative way, displacing photoreceptor nuclei at the basal side. This effect on INM led to the generation of retinal ganglion cells at the expense of inner and outer nuclear layer cells [36]. These effects are suggested to be mediated by Notch signaling, which is higher towards the apical side of the neuroepithelium, and changes in the speed of INM would alter the exposure of cells to Notch. Considering the universal role of Notch signaling in regulating the proliferative state of somatic stem cells, it is likely that many other NE-mediated effects exist linking nuclear movement to fate change decisions. In summary, the NE plays key roles during brain development and NSC maintenance, controlling the balance between NSC proliferation and differentiation.

### 2.2. Nuclear Envelope and Migration

In this section, we start by revising the contribution of lamins to the biomechanical properties of the NE during neuronal migration. We then discuss lamin-associated proteins and NPC members known to transmit mechanical forces between the cyto- and nucleo-skeleton. These processes become particularly relevant in the context of NSCs and nervous system development due to the extent of INM of NSCs as well as the remarkable migratory behavior of their neuronal progeny.

Type-A lamins increase deformation-resistant nuclear stiffness [37]. Interestingly, one of the more highly expressed microRNAs in the neural lineage, miR-9, targets the 3’ UTR of prelamin A to selectively silence lamin A isoform, but not lamin C, expression [38]. This mechanism has been proposed to protect the brain from progerin, a truncated lamin A version that causes Hutchinson–Gilford progeria syndrome, an accelerated aging disorder [38]. Thus, lack of lamin A in neural cells might grant them a deformable nucleus to navigate interstitial spaces [39]. Lamin C is also absent during development, but it is upregulated soon after birth [40]. This raises the possibility that lamin C compensates for deficits in lamin A, providing mechanical support to mature neurons once they have completed their migration. Supporting this idea, mouse embryonic fibroblasts deficient in lamin A/C exhibited nuclei with reduced stiffness, and lamin C expression by cells lacking lamin A rescued this defect [41].

Immunohistochemistry in the brain of adult mice suggested that quiescent NSCs express lamin C, but downregulate it when activated to divide [42]. Once fully differentiated, mature neurons upregulate lamin C again [42]. This observation of variable lamin C expression implies that the constitution of the NE during adult neurogenesis is dynamically reorganized. This process may be functionally relevant for the integration of adult-born neurons into existing circuits and ultimately cognitive performance and mood regulation. Future investigation is needed to elucidate these options.

Mouse lines lacking lamin B1 or B2, either systemically or forebrain-specific, exhibit a smaller cerebellum devoid of sulci and disorganized cortical layering [43,44]. Although forebrain-specific knock-out animals survive into adulthood, the cortical neurons showed migration defects, which impaired cortex layering and size [44]. The presence of nuclear blebbing in migrating neurons further suggested reduced integrity of the nuclear lamina [43,44]. It is thought that pulling forces from cytoplasmic motors deform the NE instead of translocating the nucleus towards the leading process. Importantly, some adult neurons in forebrain-specific lamin B1- or B2-deficient mice do not exhibit nuclear abnormalities, which has been related to an increase in *lmna* (namely, lamin C isoform) expression during adulthood [44]. These data point out that for migrating neurons that physiologically have virtually no lamin A/C, both lamin B1 and B2 are necessary for proper corticogenesis. Consistent with this, it was recently shown that silencing lamin B1 or B2 in migrating neurons leads to ruptures in their nuclear membrane, highlighting a role for lamins in maintaining nuclear envelope integrity during cytoskeletal pulling forces [45]. As a result of these ruptures, nuclear content is scattered into the cytoplasm, generating DNA damage and increasing cell death [45]. This suggested that the smaller brains of the knockout animals resulted not only from migration defects, but also from a decreas in cell survival as a consequence of a weakened nuclear lamina.

Filamentous lamins also serve as a scaffold for other nuclear lamina-associated proteins that aid in the mechanical regulation of nuclear dynamics. For example, Sun 1 and 2 (Sad1 and UNC-84 domain containing) interact with B type lamins, and additionally have transmembrane domains that span the inner nuclear membrane (Figure 2). Similarly, nesprin 1 and 2 are recruited to the outer nuclear membrane via a KASH domain that interacts with SUN domains at the NE lumen (Figure 2). This interaction serves as an anchor point between the nucleoskeleton and the cytoskeleton during neuronal migration and INM. Double knockout animals for either Sun 1/2 or nesprin 1/2 exhibited inverted neocortexes and enlarged ventricles [46]. Time-lapse experiments revealed that decreased expression of Sun 1/2 or nesprin 2 uncoupled the centrosomes from the nucleus during neuronal radial locomotion, preventing proper migration into the cortical plate. These mutations also reduced the number of NSCs engaged in INM and their speed towards the apical side in the VZ, which has been linked to a progressive depletion of neural progenitors [46].

Another SUN domain containing protein, klaroid, has been shown to be relevant for differentiation of the Drosophila eye. Klaroid is necessary for outer nuclear membrane localization of the KASH domain-containing protein, klarsicht. In turn, klarsicht binds to microtubules to pull the nucleus. Single- and double-knockout flies have shown that both klaroid and klarsicht are essential for apical nuclear migration. As a consequence, the eyes of these flies appear rough and irregular compared to the smooth eyes of wildtype flies [47].

Unexpectedly, the NPC is also involved in the regulation of INM of neural tube proliferating progenitors. Specifically, in mutant mice that carry a truncated version of the structural core protein, Nup133, neural progenitors fail to generate differentiated neurons [48]. It was later shown that the motor protein dynein forms complexes with Nup133 and another nucleoporin, the cytoplasmic filament RAN binding protein 2 (RanBP2). Knockdown of either complex in the developing neocortex of rats prevents both apical INM and new-born neuronal migration [49].

In sum, both in vertebrates and invertebrates, the NE hosts very well conserved machinery that transmits the mechanical forces necessary for nuclear movement and cellular positioning. As indicated in Table 1, several nervous system defects ensue upon mutations in NE components across species, affecting several progenitors and differentiated cell types.

## 3. Nuclear Envelope in Chromatin Regulation

The position of gene loci within the nucleus is dynamic and highly regulated according to cell type and developmental stage. Importantly, there is a strong correlation between the organization of the genome in three-dimensional space and the degree of gene expression [53]. In most eukaryotic cells, silenced heterochromatin localizes to the nuclear periphery, whereas actively transcribed euchromatin resides in the nuclear interior. Such distribution is thought to facilitate the necessary gene expression changes that govern developmental programs [51]. In this section, we discuss several model systems to emphasize how cell type-specific genomic architecture relies on chromatin association with different NE proteins. We then consider separately transcription inhibition and activation/bimodal roles for lamins and lamin-associated proteins first, followed by NPC members. Finally, we examine the histone modifications responsible for chromatin tethering to the NE and their effect on gene expression.

### 3.1. Nuclear Envelope and Transcriptional Inhibition

Lamins can bind to core histones, specifically H2A and H2B, and therefore serve as a docking site for chromatin at the nuclear periphery [54]. In addition, lamin-associated polypeptide 2β (Lap2β) binds to chromatin to mediate gene repression by inhibiting transcription factors such as E2F, p53 and NF-κB [55]. Thus, the genomic regions that make contact with the nuclear lamina are known as lamina-associated domains (LADs) and are transcriptionally inactive. These heterochromatic regions vary between 10 kb and 1 Mb in size and are distributed across all chromosomes. The depletion of lamin B1 leads to the detachment of LADs from the periphery, global chromatin redistribution towards the nuclear center and chromatin de-compaction in cultured cells [56]. A recent study additionally found that in the embryonic forebrain, dorsal relative to ventral neural progenitors have different LAD architectures that in turn reflect their gene expression patterns [57].

The targeting of heterochromatin to the nuclear lamina is developmentally regulated via sequential expression of lamin B receptor (Lbr) first, and lamin A/C later on [51] (Figure 2). Cells with typical peripheral heterochromatin exhibit clumps of chromatin in the nuclear center (known as “inverted nuclear architecture”) upon simultaneous Lbr and lamin A/C knockout [51]. An early study suggested that the lack of Lbr alone is sufficient to trigger inverted nucleus architecture. Several brain areas, including the cerebral cortex, hippocampus and cerebellum, of Lbr knockout mice analyzed by electron microscopy exhibited neurons with centralized heterochromatin clusters [58]. Whether or not lamin C was also hindered in these neurons due to Lbr depletion is unknown. Additionally, Lbr knockout mice displayed hydrocephaly [59], although the mechanisms leading to this defect remain elusive.

Notable exceptions to the conventional hetero- and eu-chromatin organization can be found in different systems. For example, during terminal differentiation of rods in the retina of nocturnal animals, a gradual accumulation of heterochromatin occurs in the nuclear center rather than the lamina [51]. This condensed chromatin localization leads to smaller nuclei that is thought to represent an adaptation that decreases light scattering and maximizes light sensitivity in the retinas of nocturnal mammals [60]. The loss of heterochromatin at the periphery and its accumulation in the interior depends on low levels of both Lbr and lamin A/C, since forcing the expression of Lbr in rods drives the reallocation of heterochromatin towards the NE [51].

Spatial genome reorganization at the nuclear lamina correlates with changes in gene expression [56,61,62] and, therefore, cell identity and function. Examples of this conserved mechanism have been described in several contexts for nervous system development and homeostasis, ranging from lineage commitment in culture, brain development and adulthood, and also at the level of the peripheral nervous system. For instance, during lineage commitment of embryonic stem cells from progenitor cells to astrocytes in vitro, the chromatin is dynamically reorganized closer to, or further away from, the nuclear lamina. These topological arrangements are reflected in the level of gene repression and activation, respectively [63]. Interestingly, not all genes detached from the nuclear periphery are activated in a given cell type, and at least some of them get "unfastened" for later transcription in a subsequent cellular stage [63]. Similarly, lineage-specific interactions between specific gene loci and the NE have been observed in different models. For example, during fly development, the locus of the neurogenic transcription factor *hunchback* is initially localized to the nuclear interior where it is actively transcribed, but as neurogenesis is concluded, it moves outwards to the more repressive nuclear lamina [50]. In the neuroblasts of mutant flies with decreased lamin expression, *hunchback* cannot be tethered to the nuclear periphery and, as a result, neuroblasts exhibit an extended ability to generate neurons due to prolonged expression of *hunchback* [50]. These effects are probably mediated by the association of a 250 bp intronic element within the *hunchback* locus with the polycomb repressive complex [64], but whether polycomb promotes other instances of gene-lamina tethering within the genome remains unclear. As another example, knockout of lamin B1 in mouse NSCs drastically changes the transcriptional landscape, and KEGG (Kyoto Encyclopedia of Genes and Genomes) pathway analysis of the dysregulated transcriptome in lamin B1-deficient NSCs revealed changes in the expression of genes associated with the regulation of proliferation together with transcriptional regulation and chromatin modifications [34]. Finally, at the level of the peripheral nervous system, olfactory neurons provide another example of cells with heterochromatin aggregation at the nuclear center. In these cells, only one out of hundreds of olfactory receptor genes is expressed, whereas the remaining genes converge in heterochromatic foci via a mechanism that requires low levels of Lbr [52]. Similar to the rods of nocturnal mammals, the heterochromatin in olfactory neurons does not tether with the nuclear lamina, but instead clusters in the center of the nucleus. Ectopic Lbr expression in olfactory neurons leads to heterochromatin de-compaction and co-expression of a large number of olfactory receptor genes. Given that under physiological conditions olfactory neurons in the olfactory epithelium expressing a given receptor converge within the same glomerulus in the olfactory bulb of the brain, co-expression of multiple receptors upon Lbr manipulation ultimately results in olfactory neurons targeting multiple glomeruli [52].

Components of the NPC were also shown to contribute to gene regulation and, therefore, cellular identity. Contrary to the rest of the inner surface of the nuclear membrane, NPC perforations are devoid of heterochromatin [65]. Initial studies in yeast and Drosophila reported that NPC components physically associate with genomic regions to positively or negatively regulate gene expression [66]. Although the molecular mechanisms are still unclear, it is thought that this tethering modulates the chromatin landscape, exposing or hiding genes to the transcriptional machinery. Consistent with this NPC gene silencing function, Nup153 has been shown to maintain stem cell pluripotency by binding transcriptional start sites for developmental genes, then recruits polycomb repressive complex 1 (PRC1). In mouse embryonic stem cells, knockdown of Nup153 triggered the expression of ectodermal differentiation markers and early specification of cells to the neural lineage [67].

### 3.2. Nuclear Envelope in Transcriptional Activation/Bimodal Roles

Although lamins are generally regarded as gene silencers due to their role in tethering heterochromatin to the nuclear periphery, additional unexpected roles have been described recently. Specifically, phosphorylation of serine 22 (pS22) in lamin A/C in fibroblasts induced its translocation from the nuclear periphery to the center where it was able to bind enhancers together with the transcription factor activation protein 1 (AP-1) to upregulate gene expression [68]. It will be interesting to analyze whether gene activation due to lamin A/C pS22, or other post-translational modifications, has a role in the nervous system.

Interactions between chromatin and some Nups have also been associated with transcriptional activation (Figure 2). For example, Nup210 is upregulated during neural commitment by ESCs in culture, and its knockdown blocks this lineage transition. Forced expression of Nup210 did not alter nucleocytoplasmic transport, yet it induced the expression of genes required for cell fate differentiation [69]. Similarly, chromatin binding by Nup98 varies greatly during the transition from human ESCs to neural progenitors and neurons. Although Nup98 is associated with transcriptionally active as well as inactive chromatin in ESCs, it is predominantly bound by active loci in neural progenitors and, surprisingly, Nup98 did not bind to chromatin in postmitotic neurons. In line with its association with active chromatin in neural progenitors, Nup98 overexpression increased gene transcription, particularly of neuronal differentiation genes [70]. Alternatively, the expression of a dominant negative form decreased the expression of Nup98-bound genes [70].

Accumulating evidence suggests that Nups have multiple functions in gene regulation beyond transcriptional inhibition. For example, compared to the gene silencing function of Nup153 during cell fate commitment discussed in the previous section, the same group reported that Nup93 and Nup153 can mediate both gene activation and repression. Using human cells with a bone and lung origin in culture, it was shown that Nup93 and Nup153 bind to genomic regulatory regions, named super-enhancers, that control the expression of key genes important for cell type specification [71]. Interestingly, knockdown of either Nup93 or Nup153 led to the upregulation and downregulation, respectively, of a similar number of super-enhancer-associated genes. Whether Nup93 and Nup153-mediated gene activation or repression, or a combination of both, are relevant for neural specification remains to be examined. Another study showed that nucleoporin Nup153 is an important genetic regulator in NSCs by interacting with SRY-box transcription factor 2 (Sox2) to co-regulate hundreds of genes [72]. Inhibition of Nup153 expression disrupted the genomic localization of Sox2. Interestingly, Nup153 exhibited a bimodal role in gene regulation. When bound to promoters, it activated gene transcription; conversely, when bound to transcription end sites, the complex repressed transcription. Retroviral knockdown of Nup153 in neural progenitor cells in the adult dentate gyrus induced astrogliogenesis in an otherwise neurogenic environment, underlining its relevance to cellular fate specification [72]. Thus, Nups interact with several co-factors, and their functions in gene regulation seem to be distinct depending on the context or cell type.

### 3.3. Epigenetic Modifications

As discussed above, the balance between eu- and hetero-chromatin relies on its tethering to the nuclear lamina and NPCs. Additionally, chromatin structure also correlates with histone modifications. In general, histone methylation is associated with transcriptionally silenced heterochromatin, whereas histone acetylation corresponds with transcriptionally active euchromatin [73] (Figure 2). The current view is that histone 3 lysine 9 (H3k9; and possibly also lysine 27) methylation is responsible for heterochromatin tethering to the nuclear periphery [53]. When histone deacetylase 3 (Hdac3) is bound to the transcription repressor Lap2β at the NE, it aids in the removal of gene activation landmarks [55,61].

Interestingly, although LADs are associated with gene repression, some genes within LADs are actively transcribed. Enzymatic extraction of LADs from mouse embryonic NSCs followed by H3k9me2 chromatin immunoprecipitation sequencing (ChIP-seq) revealed that the actively transcribed genes were not bound to H3k9me2. These results suggest that gene silencing within LADs can only occur in combination with this epigenetic modification [57]. Thus, gene association with various elements of the NE and histone modifications represent two complementary layers of gene regulation that operate during cell differentiation.

Additionally, chromatin rearrangements due to histone modifications are known to affect nuclear mechanical properties. For example, treatment with either histone deacetylase inhibitors or methyltransferase inhibitors to de-compact and relax the chromatin renders nuclei softer and prone to nuclear blebbing [74]. Conversely, treatment with histone demethylase inhibitors to increase compact heterochromatin results in rigid nuclei and rescues nuclear blebbing in lamin B1-knockout cells. As discussed in the previous section, changes in the biomechanical properties of the NE are often associated with impaired cell cycle exit and neuronal migration.

## 4. Nuclear Envelope in Aging

In addition to the importance of the NE in neurodevelopment and gene regulation, several reports have recently indicated that NE-associated proteins deteriorate with age and age-related neuropathology in the central nervous system [35,75,76,77,78,79,80,81]. In this section, we will discuss how, and to what extent, aging impacts NE constituents in NSCs and how damage to the NE may contribute to brain aging. Additionally, neurodegenerative diseases present with defects in nuclear integrity, nuclear compartmentalization, transport and chromatin regulation. As most of this data have been obtained in mature neurons, but might be relevant to NSCs and adult neurogenesis, we have included it in a separate section.

### 4.1. Cellular Aging

During aging, the homeostasis of cellular and molecular processes is affected (see review [82]). As a consequence, aging cells present several hallmarks that are associated with cellular dysfunction, including nucleocytoplasmic compartmentalization defects, defects in active nucleocytoplasmic transport and aging-associated transcriptional signatures. For example, a decline of nuclear lamins during aging affects the strength of the nuclear membrane in a range of tissues and cell types [83]. This causes leakage of proteins and signaling molecules into the nuclei and cytoplasm, leading to the activation of stress pathways, including oxidative damage, DNA strand breaks and differential transcription factor expression. These stress pathways damage the DNA as well as cytoplasmic organelles. Furthermore, aging- and stress-related degradation of nucleoporins changes their capacity to bind and shuttle mRNA and proteins across the nuclear membrane [75,84]. Age-related transcriptional changes lead to an induction of stress response pathway genes and a decrease in the expression of cell type-specific genes, such as those involved in synaptic transmission and calcium signaling in the brain [85]. Changes in chromatin structure, histone modifications and transcription factors play a role in triggering stress/aging pathways and downstream DNA breakdown [86,87]. All of these, can, at least in part, be regulated by NE components. This suggests the NE might be a target of aging. Some of the NE proteins, including nucleoporins and lamins, are long-lived proteins in the brain, which means they have a low turnover rate [88,89]. Furthermore, the mammalian brain only has a few known locations that possess the capacity to form new functioning neurons from remaining neural stem cell populations. Thus, these NE proteins need to be maintained over a lifetime, and there are very few possibilities to dilute or replace damaged proteins through proliferation compared to other organs or cell types. Therefore, neural cells in the brain accumulate age-associated defects in NE components over time [75,76,90]. These accumulated defects in the NE proteins may underlie age-related brain dysfunction as well as age-related neurodegenerative diseases.

### 4.2. Aging of Adult Neural Stem Cells

Recent studies using a conditional knockout animal model provided the first data on the involvement of NE proteins in the aging of adult NSC in the DG. These studies revealed that lamin B1 plays a critical role in the aging of adult NSCs [34,35]. Our study indicated that high levels of lamin B1 are essential for maintaining adult NSCs by preventing differentiation, but the enrichment of lamin B1 in adult NSCs is lost concurrently with the age-associated reduction in adult neurogenesis [34]. Precocious loss of lamin B1 induced premature neuronal differentiation in the short term, and hence, depletion of adult hippocampal neurogenesis in the long term. Furthermore, loss of lamin B1 resulted in transcriptional dysregulation and age-associated mood regulation [34]. These studies demonstrated the importance of NE components in the aging of adult NSCs.

In addition to NSC maintenance mediated by lamin B1, another role for the NE in the prevention of stem cell aging is suggested by asymmetric segregation of cellular content. With age, most somatic cells start displaying the hallmarks of aging, including ubiquitinated proteins and damaged organelles, which affect their function. A strategy that might be used by dividing cells to protect their progeny from age-associated defects is to asymmetrically segregate damaged proteins or other cellular constituents when they proliferate. Asymmetric segregation during division would allow one of the cells to remain clear of age-associated defects, whereas the other cell carries the damaged organelles and proteins. To achieve asymmetric segregation, proliferating cells use a diffusion barrier, which has been described in yeast, Drosophila and human stem cells [91,92]. In yeast, the asymmetric segregation of misfolded ER proteins in the mother cell decreases its lifespan, but the budding daughter cell has full-life expectancy. This difference in lifespan can be prevented by the knockout of diffusion barrier proteins [91], confirming the importance of asymmetric segregation to prevent aging.

The role of asymmetric segregation in the prevention of neural stem cell aging is still unclear, but Jessberger and colleagues showed that proliferating NSCs also form a diffusion barrier, which makes use of the membrane of the endoplasmic reticulum (ER). Both in vitro and ex vivo, photobleaching experiments with dividing cells demonstrated that damaged proteins could be designated to one of differentiating daughter cells [35,90]. In this way, the other proliferating daughter cell remains free of damaged proteins. However, with age, the strength of the diffusion barrier decreases in adult NSCs. Consequently, damaged proteins are distributed more symmetrically, which may contribute to the decrease in proliferation of adult NSCs. Nevertheless, the causal relationship between a failure in the asymmetric distribution of damaged proteins and the maintenance of stem cell capability still needs to be addressed [35,90].

Intriguingly, concurrent with the reduction in lamin B1 protein levels in adult NSCs, the strength of the diffusion barrier and the proliferative ability of adult NSCs decreases with age [34,35]. On the other hand, protein levels of the NE constituent SUN1, which is a member of the LINC (linker of nucleoskeleton and cytoskeleton) complex increase in aging NSCs (Figure 2) [35]. SUN1 is known to interact with lamin B1 [93], and exogenous expression of lamin B1 reduced SUN1 levels, whereas a decrease in lamin B1 leads to an increase in SUN1 in NSCs [35]. In contrast, SUN1 levels did not influence lamin B1 levels, demonstrating unidirectional regulation of SUN1 protein levels by lamin B1. Since the reduction in lamin B1 and diffusion barrier strength correlate, Bin Imtiaz and colleagues hypothesized the lamin B1-SUN1 levels influence asymmetric segregation. Indeed, photobleaching experiments confirmed the importance of high lamin B1 and low SUN1 protein levels for maintaining the strength of the ER diffusion barrier, and overexpression of lamin B1 rescued the diffusion barrier in aged adult NSCs, restoring both proliferation and neurogenesis [35]. NE components are known to integrate with the ER membrane during mitosis [94] where they could play a role in barrier strength. Yet, the exact role of lamin B1 and SUN1 in the ER diffusion barrier remains to be determined [35]. Nonetheless, these data suggest there might be a role for NE proteins in NSCs to protect against stem cell aging and defective maintenance.

### 4.3. NE-Dependent Biological Processes in Mature Neurons

In addition to stem cell maintenance and asymmetric cell division, aging affects several other biological processes that involve NE components, such as nucleocytoplasmic compartmentalization, transport and gene regulation. Here we discuss these NE-dependent biological processes, how they are affected by age and age-associated neurodegenerative diseases, and the specific roles of NE components in pathological processes. As these age-dependent impairments are mostly observed in neurons, this section focuses on the phenotypes observed in neurons.

Healthy cells maintain nucleocytoplasmic compartmentalization in order to protect the genomic material from oxidative damage and separate nuclear and cytoplasmic processes. The primary role of the NE and its components is to maintain this compartmentalization. At the same time, transport of larger molecules, including mRNA and proteins, between the two compartments is essential for proper cellular function. This nuclear-cytoplasmic transport is governed by NPC embedded in the NE as well as importins and exportins interacting with the NPC. Transcriptional signatures of aged neurons, as well as cells derived from patients or neurodegenerative disease models, differ significantly from those of young and healthy neurons [87]. Despite their importance to the maintenance of cellular processes, genes involved in nucleocytoplasmic compartmentalization and transport are downregulated with age [76]. For instance, the nuclear export supporting protein, RanBP17, is essential for the maintenance of nucleocytoplasmic compartmentalization. At the same time, a reduction in RanBP17 results in an age-associated transcriptome. It would be interesting to investigate how the reduction of RanBP17 leads to age-associated transcriptional changes in the future.

In addition to issues in nuclear transport, aberrant protein aggregation in neurons as a result of impaired proteostasis [95] is a pathological hallmark for most age-associated neurodegenerative diseases [96]. These aggregates often localize in the vicinity of the nucleus, interfering with nucleocytoplasmic transport and maintenance of the NE, which leads to nuclear blebbing, NE invaginations and eventually, cell death. Several studies have described co-aggregation of NE components with these aberrant proteins in Alzheimer’s disease [77,80], Parkinson’s disease [79,97], Huntington’s disease [98,99], amyotrophic lateral sclerosis [100] and dentatorubral-pallidoluysian atrophy [81] (see Table 2), suggesting that the NE could be a common target for neurodegenerative diseases. Future studies on the causal links between neuropathology and the effects of aggregated NE components would be of interest.

Interestingly, although many reports have identified a correlation between a decline in NE components and age-associated neurodegenerative diseases, some studies have reported an increase in NE components. For instance, a brain region-specific increase in the levels of lamin B1 is found in Huntington’s disease [105] and an extra copy of *LMNB1* gene is responsible for adult-onset autosomal dominant leukodystropy [103]. This increase leads to altered nuclear morphology and transcriptional changes as well [105]. Biomechanical changes in the NE have also been reported for adult-onset autosomal dominant leukodystropy, leading to enhanced nuclear stiffness, reduced ion channel opening capacity and reduced proliferation [104]. Thus, an appropriate balance in the composition of NE components is critical for maintaining nuclear homeostasis. Since the composition of NE components is dramatically reorganized with aging and in age-associated neurodegenerative diseases, it will be important to understand how specific ratios of NE components underlie cell type-specific functions in neurogenesis as well as neuronal function.

Collectively, emerging evidence suggests laminopathy/nucleopathy are critical mechanisms for neurodegenerative diseases. Identifying common pathways that lead to this aberrant nucleopathy could be one way to develop therapeutic treatments for neurodegenerative diseases.

## 5. Outlook on New Research Directions

After about two decades of research, our current knowledge of the NE regulation of neural stem cell fate is still limited. As we summarized above, it is clear that the NE plays pivotal roles in proliferation, migration, differentiation and gene regulation in neurogenesis, and defects in the NE lead to both developmental and age-associated diseases. This section will describe several future research directions.

First, almost 600 NE proteins have already been identified [106], but only a small fraction of them have been functionally characterized. At the same time, only a small fraction of NE components (~15%) is commonly expressed among different tissues; instead, many NE components are expressed only in the embryo or in the adult in a tissue-specific manner [107]. This spatiotemporal variability in the expression of NE components might reflect the diversity of cell type-dependent NE. The diversity of NE components could also be critical for genome organization and nuclear integrity according to cell type and developmental programs. Future research on cell type-dependent combinations of NE components, as well as uncharacterized members, will enlighten their roles in proliferation, neural cell fate decision and stem cell maintenance during neurogenesis as well as age-related pathology.

Second, the exact mechanisms that link NE impairments to brain dysfunction in aging and disease are not yet understood. Emerging evidence has uncovered several (mainly developmental) human disorders that are associated with mutations in genes encoding NE components and proteins locating and interacting at the NE (Table 3), although the underlying mechanisms remain largely elusive. Identifying these mechanisms and the specific roles of the involved NE components will support our understanding of these diseases and might open up new therapeutic avenues. At the same time, mutations in genes encoding NE components are rarely associated with age-associated diseases. Therefore, during the first decade of research, there was less attention to their involvement in adult neurogenesis, brain aging and neurodegenerative diseases. Recent studies have now shown the relevance of NE proteins in aging [76], neurodegenerative diseases [77,79,80,97,98,99] (Table 2) and more recently, adult neurogenesis [34,35]. Further studies on lamins, Nups and other NE associated proteins in adult neural stem cells could elucidate their full impact on cellular identity and cell fate maintenance.

Third, post-translational modifications of NE proteins are of interest. As mentioned, phosphorylated lamin C plays a role in active gene expression in fibroblasts [68], whereas non-modified lamin C is rather involved in gene silencing. Since lamin C protein levels differ depending on differentiation stage—high in both NSC and mature neurons, but lower in differentiating and migrating neurons [42]—phosphorylation or acetylation [128] of lamin C could be of relevance for gene regulation during neurogenesis. On top of that, this report prompts us to hypothesize a wide range of possible gene regulatory effects for post-translational modifications on NE components that might influence gene expression, mitosis or migration in different cell types and contexts.

Finally, another interesting phenomenon is nuclear invagination, which is a common feature in aging, tumors and many neurodegenerative diseases [129] that is also observed in physiological conditions. The mechanisms behind nuclear invagination in different diseases have been associated with a reduction or dysfunction of lamin B1 [77,81,130]. However, the biological roles of nuclear invagination are still unclear. Interestingly, hematopoietic stem cells also present with nuclear invaginations, and the findings imply a physiological role for nuclear invagination in fate specification [131]. In addition, a subgroup of post-natal NSCs in the subventricular zone were shown to present with large nuclear invaginations termed envelope-limited chromatin sheets. The presence of these invaginations was linked to their quiescent state, and therefore deemed a physiological presentation of the nucleus [132]. The physiological roles of nuclear invagination in both embryonic and adult NSC fate specification will be most interesting to investigate.

## Figures and Tables

**Figure 1 cells-11-01761-f001:**
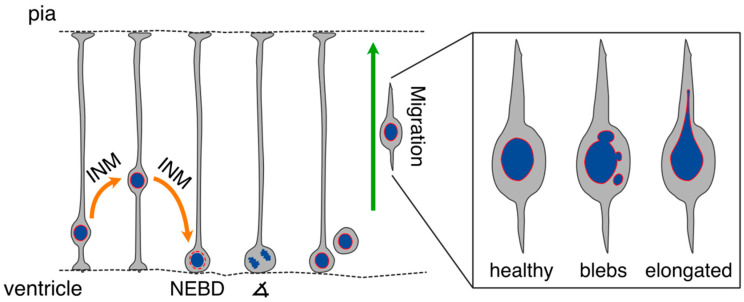
Scheme of the embryonic cortex illustrating essential steps during neurogenesis that are known to be regulated by NE proteins. NSCs make contact with both apical and basal lamina, and their soma undergoes interkinetic nuclear migration (INM) following the cell cycle. Mitoses occur at the ventricular surface of the cortex, beginning with nuclear envelope breakdown (NEBD). Mitotic spindle orientation (∡) determines whether a cell division will be symmetric (producing two identical daughter cells) or asymmetric (producing one NSC and another daughter with a more restricted cell fate). Once mitosis is complete, new-born neurons migrate to their final position within the cortical plate. Intermediate basal progenitors were omitted for simplicity. Mutations in NE components often lead to nuclear rupture or deformation in migrating neurons, causing blebs and elongations.

**Figure 2 cells-11-01761-f002:**
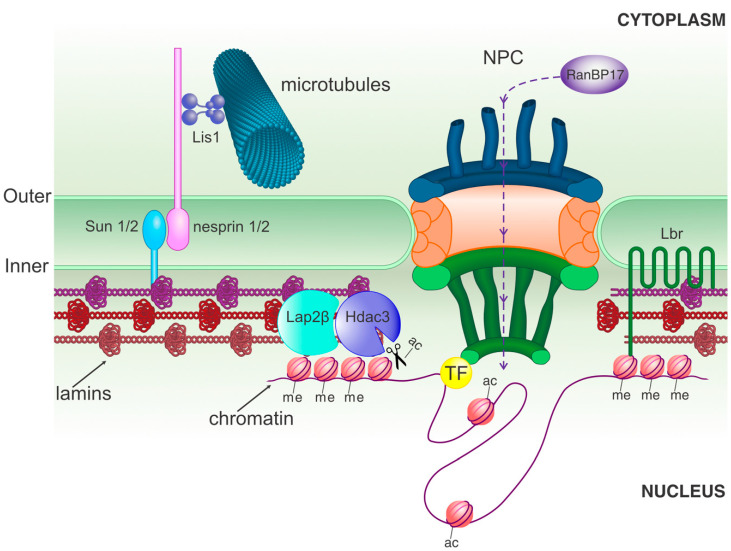
Scheme of the nuclear envelope and the associated components discussed in the main text. NPC, nuclear pore complex; TF, transcription factor; me, methylation; ac, acetylation.

**Table 1 cells-11-01761-t001:** Nuclear envelope components known to affect neural development.

NE Component	Species	Cell Type	Phenotype	Reference
**Klaroid or klarsicht**	Drosophila melanogaster	Photoreceptor	Rough eyes	[47]
**Lamin**	Drosophila melanogaster	Neuroblast	Extended neurogenesis	[50]
**Lamin B1 or B2**	Mus musculus	Radial glial cells and neurons	Disorganized cortex, smaller cerebellum	[31,43,44]
**Lbr**	Nocturnal mammals	Rods	Heterochromatin inversion	[51]
**Lbr**	Mus musculus	Olfactory neurons	Co-expression of multiple olfactory genes, and glomeruli mistargeting	[52]
**Lis1**	Mus musculus	Radial glial cells	Lissencephalic brain	[30]
**Nesprin 2**	Danio rerio	Retinal ganglion cells	Smaller eyes	[36]
**Nup133**	Mus musculus, Rattus norvegicus	Radial glial cells and neurons	Exencephalic neural tube	[48,49]
**Sun 1/2 or nesprin 1/2**	Mus musculus	Radial glial cells and neurons	Inverted cortex and enlarged ventricles	[46]

**Table 2 cells-11-01761-t002:** Changes in NE components (bold letter) upon aging and in neurodegenerative diseases and the resulting phenotypes. iN, neurons directly reprogrammed from human-derived fibroblasts; mHtt, mutant Huntingtin; HD, Huntington’s disease; ALS, amyotrophic lateral sclerosis; FTD, frontotemporal dementia; pTau, hyperphosphorylated tau; AD, Alzheimer’s disease; CA1, cornu ammunis 1; PKC, protein kinase C; polyQ, poly-glutamine; DRPLA, dentatorubral-pallidoluysian atrophy; ADLD, adult-onset autosomal dominant leukodystropy; LRRK2, leucine-rich repeat kinase 2; PD, Parkinson’s disease.

NE Component	Species	Cell Type	Change from Physiological State	NE Component Changes	Phenotype	Refs.
**RanBP17**	Homo sapiens	iN	Aging	Age-dependent transcription level decrease	Decline in nucleocytoplasmic compartmentalization, age-associated transcriptome	[76]
**RanGAP1, Gle1, Nup62**	Mus musculus	Cortical neurons	mHtt (HD)	Interaction with mHtt, subsequent accumulation and perinuclear mislocalization	Disrupted nucleocytoplasmic transport, leaky and compromised nuclear pore	[98,99]
**RanGAP1, Nup107, Nup205**	Homo sapiens	Neurons from patient-derived iPSC	C9orf72 mutation (ALS/FTD)	Accumulation and perinuclear mislocalization	Nucleocytoplasmic transport defects	[100]
**Nup98**	Homo sapiens	Hippocampal neurons	pTau (AD)	Interacts with pTau, accumulation and perinuclear mislocalization	Hampers nucleocytoplasmic transport, disrupts NPC distribution, accelerates and stabilizes pTau aggregation	[80]
**Msp300 and koi (homologs of human Nesprin and SUN1)**	Drosophila melanogaster	Cortical neurons	pTau (AD)	Interaction with pTau subsequent induction filamentous actin, mislocalization in foci and nuclear blebs	Lamin B loss, nuclear invaginations, heterochromatin relaxation and DNA damage	[77]
**Lamin B1**	Homo sapiens	Frontal cortex neurons	pTau (AD)	Reduction in protein level	Nuclear invagination, heterochromatin relaxation	[77]
**Lamin B1**	Homo sapiens	Neuroblastoma cells	Aβ42 (AD)	Protein cleavage after Aβ42 dependent release of Cathepsin L from lysosomes	Nuclear invaginations	[101]
**Lamin B1**	Mus musculus	Hippocampal CA1 & striatal neurons	Downregulated PKCδ kinase (HD)	Protein level increase up to 4× wildtype level	Altered nuclear morphology, transcriptional changes	[102]
**Lamin B1**	Homo sapiens	Neurons/patient-derived fibroblasts	PolyQ ataxin (DRPLA)	Protein level decrease and co-localization with ataxin aggregates	Cytoplasmic localization and degradation, nuclear invagination	[81]
**Lamin B1**	Homo sapiens	HEK293	ADLD	Extra copy of *LMNB1*	Altered nuclear morphology	[103]
**Lamin B1**	Homo sapiens	Patient-derived fibroblasts	ADLD	Extra copy of *LMNB1*	Enhanced nuclear stiffness, reduced ion channel opening capacity, reduced proliferation	[104]
**Lamin B1, B2, C**	Homo sapiens	Neurons from patient-derived iPSC, brain sections	*LRRK2* G2019S mutation (PD)	Abolished interaction between LRRK2 and lamins	Decline in nucleocytoplasmic compartmentalization, nuclear envelope disorganization	[79,97]

**Table 3 cells-11-01761-t003:** List of human disorders and conditions associated with mutations in NE components (bold letter). Note that some NE proteins have additional sub-cellular localizations, and whether the reported disorder is due to the role of the gene product in the NE or elsewhere remains to be clarified. ADHD, attention deficit hyperactivity disorder; ANE1, acute necrotizing encephalopathy type 1; ASD, autism spectrum disorder; B, bipolar disorder; CA, cerebellar ataxia; D, depression; EPI, epilepsy; IBSN, infantile bilateral striatal necrosis; ID, intellectual disability; SCZ, schizophrenia.

Gene	Disorder/Condition	Localization in Addition to the NE	References
** *AAAS* **	Triple-A syndrome		[108]
** *DMPK* **	ASD	mitochondria, cytoplasm	[109]
** *DST* **	SCZ, ASD	cytoplasm	[110]
** *ITSN1* **	ASD, ID, EPI	vesicles, cytoplasm	[111]
** *NUP62* **	IBSN		[112]
** *NUP85* **	Microcephaly		[113]
** *NUP107* **	Microcephaly		[114]
** *NUP133* **	ASD, Microcephaly		[115,116]
** *NUP155* **	ASD		[117]
** *NUP214* **	Microcephaly		[118]
** *RANBP2* **	ANE1		[119]
** *SPAST* **	ASD	ER, cytoplasm	[120]
** *SYNE1* **	ASD, CA, B, D	nucleus, cytoplasm	[121,122,123]
** *SYNE2* **	ASD, ID	nucleus, cytoplasm	[124]
** *TERB2* **	ASD		[125]
** *WDFY3* **	ASD, ADHD	vesicles, cytoplasm	[126]
** *XPO1* **	ASD	nucleoplasm, cytoplasm	[127]

## Data Availability

Not applicable.

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
