# Peer review of "A Nuclear Belt Fastens on Neural Cell Fate"

_cells, 2022, doi:10.3390/cells11111761_

Round 1

Reviewer 1 Report

This work from Mestres et al. reviews elegantly a very interesting topic, as it is the contribution of the nuclear envelop during neural development and its alterations during aging.  The manuscript is well written and contains an extensive revision of the bibliography associated with the subject of the study, however, I have a couple of concerns that I believe the authors should address to improve the manuscript:

-          The first one is related with the practical absence of figures in the manuscript that makes the text quite dense and sometimes hard the follow. From my point of view the inclusion of schemes in some of the sections, for example when explaining roles in nuclear division or neuronal migration would improve the readability of the manuscript.

-          While the sections related with the role of the nuclear envelop during nervous system development and during aging are expected from the introduction, the second section related with chromatin regulation seem a bit loose in between these other sections. Authors should introduce this part in the initial introduction and improve the transitions between each of the parts of the manuscript. 

Author Response

We thank you for your time reviewing our manuscript and your constructive criticism. In the next point-by-point responses, we explain the introduced changes to address your concerns.

Reviewer 2 Report

In this review the authors identified an emerging field in the study of mechanisms underlying neurodevelopment and neurodegeneration, by dissecting the role of NE in this contexts. The review is well written, and all the different aspects well described and discussed. I do not have major comments. If the lenght of the manuscript wont be a limitation, I would suggest to add a paragraph on human disorders due to mutations of genes belonging to the NE family. This will be interesting and to increase the importance of this review. For example there are emerging studies showing the existence of mutations in nesprin genes in patients with autism (see as an example PMID: 34573277).

Author Response

(The authors gave the same response as above.)
